# Differences in Carbon Sequestration Ability of Diverse Tartary Buckwheat Genotypes in Barren Soil Caused by Microbial Action

**DOI:** 10.3390/ijerph20020959

**Published:** 2023-01-05

**Authors:** Wei Chen, Zhiwei Zhang, Congjian Sun

**Affiliations:** School of Geographical Science, Shanxi Normal University, Taiyuan 030031, China

**Keywords:** tartary buckwheat, C-transformation enzymes, carbon sequestration, ozone sterilization

## Abstract

Planting plants to increase soil carbon input has been widely used to achieve carbon neutrality goals. Tartary buckwheat not only has good barren tolerance but is also rich in nutrients and very suitable for planting in barren areas. However, the effects of different genotypes of Tartary buckwheat roots and rhizosphere microorganisms on soil carbon input are still unclear. In this study, ozone sterilization was used to distinguish the sources of soil organic acids and C-transforming enzymes, and the contribution of root and rhizosphere microorganisms to soil carbon storage during the growth period of two genotypes of tartary buckwheat was studied separately to screen suitable varieties. Through the analysis of the experimental results, the conclusions are as follows: (1) The roots of Diqing tartary buckwheat have stronger carbon sequestration ability in a barren environment than Heifeng, and the microorganisms in Diqing tartary buckwheat soil will also increase soil carbon input. Therefore, Diqing tartary buckwheat is more suitable for carbon sequestration than Heifeng tartary buckwheat in barren soil areas. (2) In the absence of microorganisms, the rhizosphere soil of tartary buckwheat can regulate the storage of soil organic carbon by secreting extracellular enzymes and organic acids. (3) The structural equation model showed that to promote carbon sequestration, Heifeng tartary buckwheat needed to inhibit microbial action when planted in the barren area of Loess Plateau, while Diqing tartary buckwheat needed to use microbial-promoting agents. Adaptive strategies should focus more on cultivar selection to retain carbon in soil and to assure the tolerance of fineness in the future.

## 1. Introduction

The improvement of global industrialization and the overexploitation of fossil fuels have caused global warming and a series of environmental degradation problems [1]. Reducing carbon emissions and achieving carbon neutrality have become the common goals of countries in the world [2,3]. The soil carbon pool is the largest carbon pool in terrestrial ecosystems, and its carbon storage in temperate regions is three times more than that of plants. Therefore, increasing carbon storage in soil is an effective way to alleviate climate change [4,5]. Plants play an important role in soil carbon storage. While consuming and utilizing carbon during growth, they also input a large number of carbon-containing compounds into the soil. With the improvement in people’s quality of life and the demand for physical health, a large number of nourishing food has been sought after. Tartary buckwheat is a dicotyledonous plant of the genus Fagopyrum of the family Polygonaceae, mainly cultivated in Asia (China, India, etc.), and China’s tartary buckwheat production areas are mainly concentrated in the southwest mountain and loess plateau [6,7]. Tartary buckwheat is rich in nutrition, not only rich in protein, but also contains higher flavonoids. Its unique chemical composition makes it commonly used in adjuvant therapy for certain chronic diseases, such as diabetes, hypertension, etc. [8]. At the same time, tartary buckwheat also has a strong tolerance to barren soil and high yield, and low requirements for planting environment [9]. Therefore, it is very suitable for the promotion of carbon storage engineering plants selected as barren areas.

Compared to aboveground parts, belowground parts of plants play a greater role in soil carbon input. Gale et al. found that up to 75% of soil carbon input comes from belowground parts of plants [10], while Barber and Martin et al. found that 1-year-old grains input 30–50% of the carbon fixed by photosynthesis into rhizosphere soil [11]. The organic acid is an important component of active organic carbon in the soil, which is mainly derived from plant root exudates and microbial metabolic secretion [12]. Studies have shown that the type and content of organic acids affect the priming effect, which in turn changes soil carbon storage [13]. Soil C-transforming enzymes are a tool for plants and microorganisms to absorb and utilize carbon. The changes in different soil enzyme activities can reflect the differences in organisms’ utilization of carbon sources [14]. As the gatekeeper of soil-atmosphere carbon exchange, soil microorganisms are more sensitive to changes in soil nutrients. In nutrient-poor areas, microbial activity is inhibited, while in nutrient-rich areas, microbial metabolism and respiration are enhanced, which will increase the consumption of soil carbon [15]. However, plants secrete more root exudates to use microorganisms [16], and there is also a phenomenon that oligotrophic microorganisms use complex carbon sources to increase the loss of stable carbon sources [17]. Therefore, explaining the contribution of microorganisms and plants to soil carbon storage is necessary.

The Loess Plateau is one of the most severely eroded areas in China. ‘Nitrogen deficiency, phosphorus deficiency, and potassium deficiency’ has become the main characteristic of the local soil [18]. Lack of nutrients in the soil and ease to drain makes the Loess Plateau widespread over-fertilization, resulting in environmental pollution and waste that has attracted people and government attention [19]. Compared with the main crop corn, tartary buckwheat needs less fertilizer application, and tartary buckwheat has economic value, so it is suitable for local promotion and planting [20,21]. Studies have shown that different genotypes of tartary buckwheat have different root activity and underground biomass in poor soil [22]. Therefore, selecting suitable tartary buckwheat varieties is very important. Based on this, this study selected two tartary buckwheat varieties: Heifeng (a low nutrient-tolerance variety) and Diqing (a low nutrient-sensitive variety) and carried out ozone sterilization pot experiments. The plants and microorganisms were studied separately to determine the contribution of plants and microorganisms to soil carbon storage. At the same time, the tartary buckwheat genotypes more suitable for planting in the Loess Plateau were screened to provide new ideas for local ecological construction. We hypothesized that: (1) Tartary buckwheat varieties with strong barren tolerance will also have stronger carbon sequestration capacity. (2) The rhizosphere soil of Tartary buckwheat can regulate the storage of soil organic carbon by secreting soil extracellular enzymes and organic acids in the absence of microorganisms.

## 2. Materials and Methods

### 2.1. Site Description and Sample Preparation

The study was carried out in a greenhouse in Linfen city, Shanxi province, China (111°31′ N, 36°05′ E). This city has a temperate monsoon climate, and the mean annual precipitation and temperature are 550 mm and 14.3 °C, respectively. The experimental soil was collected from barren surface soil (0–15 cm) in a 3-year fallow arable land at Xiangning soil and water conservation monitoring station. The soil classified as Chromic Cambisols according to World Reference Base for Soil Resources (WRB) classification.

The greenhouse pot experiment was started in 2017, with tartary buckwheat transplanted in early May and harvested in mid-August. Two tartary buckwheat cultivars were selected: Heifeng (HF), A genotype screened for poor tolerance to low soil nutrients; and Diqing (DQ), a genotype screened for high tolerance to low soil nutrients [22]. A total of 16 pots were used in this study. Each pot was about 21 cm in diameter and 20 cm in height, and each pot contained 10 kg of experimental soil and 16 carefully selected tartary buckwheat seeds.

Two kinds of tartary buckwheat seeds were watered with 400 mL 1 mg L^−1^ ozone water per day after sowing (8 May) as germicidal treatment (O3), while another group of tartary buckwheat seeds was watered with 400 mL of deionized water per day as control (CK). No fertilizer was applied in the experiment.

In this study, soil rhizosphere samples were collected at the seedling stage (25 June), flower stage (25 July) and maturity stage (31 August) of tartary buckwheat. The soil samples at different stages were stored separately for experiment and analysis. Every treatment data was measured by select four pots. Three tartary buckwheat plants were randomly selected from each pot and taken out completely. By shaking off the non-rhizosphere soil on the root system, the soil attached to the root surface was 1–2 mm thick, and then immediately brought back to the laboratory to brush the surface soil with a brush and stored in the experimental bag. Each experimental bag was composed of three strains of rhizosphere soil from the same pot and fully mixed. The weight of soil rhizosphere samples in each experimental bag was at least 15 g. The soil rhizosphere samples were kept at 4 °C for chemical analyses and −20 °C for enzyme activity and organic acid content analysis.

### 2.2. Analysis of Soil Properties

Soil moisture (SWC) was determined by drying at 105 °C for 24 h. Soil pH was determined in a 1:2.5 (soil:water) solution (*w*/*v*). Soil organic C (SOC) was detected with the water and hot potassium dichromate oxidation method [23].

### 2.3. Analysis of Carbon Transformation Enzyme Activities

β-glucosidase activity was measured by the method of Eivazi and Tabatabai [24], using the substrate analogue para-nitrophenyl-β-d-glucopyranoside (pNPG), Moist soil (1.00 g) was weighed into screw-cap glass test tubes and incubated for 1 h in a water bath at 37 °C with 4 mL of 0.05 M modified universal buffer (pH 6.0) and 1 mL of 25 mM pNPG (5 mM final concentration) dissolved in buffer. The reaction was terminated by adding 1 mL of 0.5 M CaCl_2_ and 4 mL of 0.2 M Tris–hydroxymethyl (aminomethane), adjusted to pH 12 with NaOH. The mixture was centrifuged for 10 min at 1500× *g* and the absorbance was measured at 410 nm. The β-glucosidase activity was expressed as mg glucose (kg soil h^−1^) [25].

Sucrase and cellulase activities were determined according to the method of Guan [26], Sucrase activity was assayed using sucrose solution as substrate. After incubating at 37 °C for 24 h, the filtrate was boiled in a water bath for 5 min with 3 mL 3,5-dinitrosalicylic acid (DNS). The absorbance of the reducing sugars was measured at a 508 nm wavelength, and sucrase activity was expressed as mg glucose (g soil 24 h^−1^).

Cellulase activity was assayed using carboxymethyl cellulose as substrate. After incubating at 37 °C for 72 h, the filtrate was boiled in a water bath for 5 min with 3 mL 3, 5-dinitrosalicylic acid (DNS). The absorbance of the reducing sugars was measured at a 540 nm wavelength and cellulase activity was expressed as mg glucose (g soil 72 h^−1^).

### 2.4. Analysis of Organic Acids Content

The concentrations of organic acids were measured by high-performance liquid chromatography (HPLC). A total of 2.5 g tartary buckwheat rhizosphere soil samples were extracted by 5 mL of H_3_PO_4_ (0.1%) solution. Then the mixture was centrifuged at 5000 g for 5 min after shaking for 1 min. The supernatant was filtered and stored at 4 °C for further analysis. The supernatant was measured by HPLC apparatus with a SWELL Chromplus C18 column (250 mm × 4.6 mm × 5 mm). The temperature and wavelength were at 35 °C and 210 nm. The mobile phase consisted of 25 mM H_3_PO_4_ (pH 2.1) and C_2_H_3_N (98:2, *v*/*v*) at a flow rate of 1 mL/min [27]. The tested organic acids content includes malonic acid (Pda), propionic acid (Pa), formic acid (Fa), oxalic acid (Oa), tartaric acid (Ta), malic acid (Ma), lactic acid (La), acetic acid (Aa).

### 2.5. Statistical Analyses

A general linear model analysis of variance designed for split-plot design was performed to test whether the sterilization, period, and variety were significant for the measured indicators. Period, sterilization, and variety were fixed as indicators, and repetition was a random factor. When the treatment effect was significant, the independent sample *T*-test was used to analyze whether the difference of varieties was significant, and the analysis of variance (ANOVA) was used to analyze whether the difference between the same variety was significant under sterilization treatments and different growth stages. All statistical analyses were performed by SPSS statistical software (SPSS Inc., Chicago, IL, USA). Differences at *p* < 0.05 level are considered to be statistically significant. Redundancy analysis (RDA) was conducted to determine the influence of environmental factors on the soil C-transforming enzymes. RDA made into a figure using cannco 4.5. Ratios of CK to O3 treatment data were made into a scatter plot. The data was divided into two parts by the dashed line of y = x. The closer the point to the upper left corner, the stronger the microbial carbon sequestration ability was, and the closer the point to the lower right corner, the stronger the plant carbon sequestration ability was. Origin software was a tool to draw the histogram and show the differences between the different treatments in letters. All sample data were processed by the ln function in Excel, and scatter plots and correlation analysis combination plots were drawn by the ggplot package in R language. The structural equation model (SEM) was used to evaluate the effect of sterilization treatment and differences on soil carbon sequestration, and each index’s direct, indirect, and total effects on soil organic carbon were drawn into a table. In the AMOS 24.0 environment, the maximum likelihood estimation method was used for model analysis, and the overall fitting degree of the model was evaluated by the chi-square test (*p* > 0.05), root means square of the approximation error, Akaike information criterion, and goodness of fit index.

## 3. Results

### 3.1. Soil pH, SWC, and SOC

The soil samples’ physicochemical data before the pot experiment follows: 1.33 g kg^−1^ total organic carbon, 0.222 g kg^−1^ total nitrogen, 3.69 ug g^−1^ available phosphorus, 100.12 mg kg^−1^ available potassium, pH 8.0. The univariate analysis of variance results showed that pH, SWC, and SOC were all significantly (*p* < 0.01) changed by period, treatments (except SWC), cultivar, and the interaction between period, treatment, and cultivar (Figure 1). At the seedling and flowering stage, the SWC of HF was higher than that of DQ, and the SWC of HF decreased at the seeding and flowing stage and increased at the mature stage under sterilization treatment. Similarly, pH decreased at the seedling stage and flowering period under sterilization treatment, while it did not significantly change at CK from seedling to flowering. SOC had no significant difference between HF and DQ under the sterilization treatment, but increased/decreased trends were found in HF/DQ at the flowering period. At the same time, SOC significantly decreased in HF and increased in DQ from flowering to maturity under the sterilization treatment (*p* < 0.05). It was worth mentioning that SOC did not change at the mature stage under sterilization treatment. A significant difference was found in SOC between HF and DQ at the seedling and flowering under the sterilization treatment. Interestingly, the highest SOC appeared at the flowering stage of HF under sterilized treatment and was 41.62% higher than that of DQ.

### 3.2. C-Transforming Enzymes in Soil

The univariate analysis of variance results showed that C-transforming enzymes were all significantly (*p* < 0.01) changed by period, treatments, and the interaction between period, treatment, and cultivar (Figure 2). However, there were no significantly changed in the cultivar of SC and CE. Sterilization significantly changed BG activity in the soil at all stages of tartary buckwheat, they showed that the BG activity of HF was an increase but DQ decreased.

Meanwhile, sterilization significantly increased the SC activities of HF and DQ at seedling and flowering stages, and the CE activities of HF and DQ were also observed to significantly increased (*p* < 0.05) at seedling and maturity stages. Interestingly, the lowest value of HF and the highest value of DQ in BG activity appeared in the CK of the flowering period. However, HF increased by 505.46% and DQ decreased by 50.46% after sterilization. The highest SC activity of HF and DQ appeared in the seedling and flower stage of sterilization treatment. As well, they respectively increased by 127.11%, and 238.54% when compared with CK. The highest CE activity of DQ appeared in the mature sterilization treatment, which was 146.63% higher than that of CK. According to the change of period, the activity of BG had no significant changes under the treatment of sterilization in DQ, yet a significant decrease was found in HF. There was no significant difference in BG activity between HF and DQ under CK treatment at the seedling stage. Compared with CK at the flowering stage, a 78.26% decrease and a 55.56% increase were found under HF and DQ of BG, respectively. It was worth noting that the CE activity of HF and DQ decreased during the mature period in CK, but the sterilization made the CE increase significantly (*p* < 0.05).

### 3.3. Organic Acids

Fa, La and Aa were not detected in all samples, and the univariate analysis of variance results showed that the other organic acids were all significantly (*p* < 0.01) changed by period, treatments, cultivar, and the interaction between period, treatment, and cultivar (Figure 3). Ma and Pa were detected only in the flowering period, which could be found in HF and DQ under the treatment of sterilization and CK treatment, respectively. Oawas not detected in the rhizosphere soil of DQ at the seedling stage and was found significantly higher than that of HF at the flowering stage under CK (*p* < 0.05). Oa was significantly decreased in HF and DQ at the flowering stage under sterilization treatment. It was also detected in DQ at maturity. Tawas a significant increase in HF from the seedling stage to the flowering stage under sterilization treatment. It was worth noting that during the whole growth period of the two varieties, the highest Ta content in DQ was found under CK treatment at the flowering stage and was 332.31% higher than HF under the same conditions, while it decrease 90.63% under O3. The highest content of Tain HF appeared in the same period of sterilization treatment, but it was 54.38% lower than the highest value of DQ. Ma could not be detected in DQ at the seedling stage and flower stage under sterilized treatment, while Mawas detected in HF soil which was not detected in CK after sterilization, and it was the highest in all treatments under different periods. Under the sterilization treatment, DQ was significantly higher than that of HF.

### 3.4. Correlations of Organic Acids, Enzymes and the Environmental Variables

The redundancy analysis (RDA) showed the relationship between C-transforming enzymes and environmental factors (Figure 4a–c). It showed that at the seedling stage (a) RDA1 distinguished HF and DQ samples under sterilization treatment (81.3%), and RDA2 distinguished the two treatment samples of tartary buckwheat (13.5%). At the seeding stage, Ta was the closest positively correlated of organic acids with SC. The organic acid most closely related to BGwas Ma, and it was a positive relationship. Among the organic acids, Oa was most closely related to CE and was negatively correlated. SOC was positively correlated with BG and negatively correlated with CE. There was a negative correlation between pH and three C-transforming enzymestC-transforming enzymes.

At the flowering stage (b) RDA1 distinguished the samples of the two varieties (57.0%), and RDA2 distinguished the samples of the two treatments (41.2%). BG was positively correlated with all detected organic acids, while SC and CE were negatively correlated with them, and CE was more closely related to organic acids. Consistent with the seedling stage, pH was positively correlated with C-transforming enzymes, but it was worth noting that SWC was negatively correlated with BG.

At maturity (c) RDA1 distinguished the two varieties treated with CK (65.7%). At the mature stage, Ta was the organic acid most closely related to BG and CE, while Ma is most closely related to CE and is positively correlated. Different from the seedling and flowering stages, pH was positively correlated with CE at the maturity stage.

Pearson correlation analysis of HF and DQ separately yielded Figure 5a,b from which it could be seen that Pda and Pa had one of the highest correlations of two cultivars. Under HF, Pda, Pa, Ta, and Ma were significantly correlated with C-transforming enzymes, and all of them were positively correlated. Under DQ, all organic acids were significantly positively correlated with C-transforming enzymes BG. However, SC was significantly negatively correlated with all organic acids. Ma was significantly positively correlated with SC of HF but negatively correlated with SC in DQ (*p* < 0.001). The most significant C-transforming enzymes affecting SOC in HF was cellulase (0.65), which was positively correlated.

Pda, Pa and Ta in HF were significantly positively correlated with SOC. All the organic acids detected in DQ showed a significant positive correlation with SOC. The organic acids with the highest correlation with SOC were Pda and Ma of HF and DQ, respectively. pH was significantly negatively correlated with all organic acids (except Ma) and SC in HF, while only Ma was significantly positively correlated with pH in DQ. Only in DQ, pH showed a significant positive correlation with SOC. SWC occurs in HF and was negatively correlated.

### 3.5. Effects of Sterilization and Cultivar on Soil Carbon Sequestration

At the seedling stage (Figure 6a) Ta of HF and DQ was the most significant in promoting microbial carbon sequestration, while Ma of DQ was more advantageous than other indexes in promoting plant carbon sequestration. At the flowering stage (Figure 6b), Oa and Ta of DQ had the most significant effect on promoting microbial carbon sequestration, while BG of HF had more advantages than other indicators in promoting plant carbon sequestration. At maturity stage (Figure 6c) Oa of DQ had the most obvious effect on promoting microbial carbon sequestration, while Ta of HF had the most obvious effect on plant carbon sequestration. It was worth mentioning that BG, Oa, and Ta of HF all tend to promote microbial carbon sequestration.

To consider the contribution of varieties and sterilization treatments to organic carbon storage, the contribution of varieties and sterilization treatments to carbon sequestration during the whole growth period of tartary buckwheat was analyzed by SEM analysis (Figure 7). The SEM revealed that the predictors explained 81.5% of the variation in C storage indifferent cultivars and 86.3% in germicidal treatment. The difference in a variety had a positive effect on organic acid and C-transforming enzymes, and C-transforming enzymes promoted the accumulation of organic carbon. However, the increase of organic acid not only reduced the accumulation of organic carbon but also had a negative effect on C-transforming enzymes activity and pH. Although pH value directly promoted organic carbon sequestration, it had a negative effect on C-transforming enzyme activity. SWC, variety inhibits its growth and it also had a negative effect on SOC.

Sterilization treatment promoted the sequestration of SOC in the soil through the positive effect on C-transforming enzymes, and also further promoted the sequestration of organic carbon by affecting organic acid content. Organic acids had a negative effect on organic carbon and pH, but have a positive effect on SWC. The increase in pH promoted the sequestration of organic carbon, while the increase in SWC was not conducive to sequestration. Therefore, the negative effect of sterilization on organic acids will promote the sequestration of organic carbon. The structural equation model (Table 1 and Table 2) shows the influence of each index on SOC under the change of variety and sterilization treatment. Table 1 and Table 2 showed that the most direct influence on SOC under variety change and sterilization treatment was pH, and the most indirect influence was Ta which had a negative influence. In terms of total impact, the greatest impact on SOC after the variety change was pH, and the smallest was Oa.

## 4. Discussion

### 4.1. Response of pH, Water Content, and Soil Organic Carbon to Sterilization

pH is an important environmental indicator affecting soil plants and soil microorganisms and is also considered to be an important driving factor that can affect soil carbon storage [28]. In general, microorganisms can stimulate the production of plant root exudates by secreting growth-stimulating hormones and root colonization, thereby affecting the pH of rhizosphere soil. For example, Munir J et al. found that the pH of wheat rhizosphere soil decreased after arbuscular mycorrhizal colonization without adding phosphorus [29], but the pH of rhizosphere soil of two kinds of tartary buckwheat decreased significantly at the seedling stage and flower stage after sterilization in the experiment, which may be attributed to the low colonization level of bacteria and fungi in tartary buckwheat roots. At the same time, plants can regulate pH by secreting organic acids, etc. [30], while only a small number of the eight kinds of aliphatic organic acids measured in this paper showed that growth phenomenon under sterilization conditions, it was inferred that sterilization treatment may lead to the secretion of more aromatic organic acids which were difficult to be decomposed by microorganisms, thereby reducing the pH value and these organic acids also enhance the soil carbon content and stability. As the main organ for plants to absorb and utilize water, roots are not only extremely sensitive to soil moisture but also enhance soil water-holding capacity by regulating soil structure through root exudates, thereby affecting soil carbon storage and utilization [31]. Soil microorganisms can also change soil water holding capacity and increase soil water content by their secretion and stimulating plant root growth [32]. HF and DQ showed significant differences in soil water content in this experiment. Sterilization treatment led to a decrease in soil water content in the rhizosphere of HF tartary buckwheat. It was precise because sterilization caused the soil to lack the stimulation of microbial physiological activities so the soil water holding capacity was not improved. The soil water content of DQ did not change significantly after sterilization, which showed that the root system of DQ had a stronger ability to regulate soil water holding capacity.

As a key index to measure soil carbon storage, organic carbon is also particularly important in the metabolism of plants and microorganisms. Microorganisms can achieve organic carbon sequestration by decomposing and transforming organic carbon, and microorganisms can also consume soil organic carbon storage by respiration [33]. Sterilization led to a significant increase in SOC in HF during the flowering period but reduced the SOC content of DQ, which indicated that the rhizosphere microorganisms of HF consumed a large amount of organic carbon in the soil during the flowering period, while the rhizosphere microorganisms of DQ promoted the accumulation of SOC. The change in organic carbon in the period was also worthy of attention. In the early stage of plant growth, a large amount of carbon was needed to promote the formation of plant roots and stems, while in the flowering stage, the demand for nitrogen, phosphorus, and other nutrients was sought. At the same time, a large number of carbon-containing compounds are released into the soil because the root system of the flowering stage becomes more developed than the seedling stage. Therefore, the content of organic carbon in the rhizosphere of tartary buckwheat increased at the flowering stage. However, the carbon will be consumed in large quantities at the maturity stage due to the aging of the root activity. The soil organic carbon pool will be supplemented by microbial decomposition of plant residues after plant litter [34]. In the experiment, the SOC of tartary buckwheat also showed a trend of increasing first and then decreasing with plant growth. However, under sterilization conditions, the loss of organic carbon in HF from flowering to maturity was significantly greater than that in DQ. At the same time, through the comparison of SOC content between the seedling stage and maturity stage, it could be found that the organic carbon storage in DQ was greater than that in HF during the whole growth period, which indicates that DQ roots have stronger carbon sequestration potential than HF.

### 4.2. Response of Soil Organic Acid to Sterilization

Root exudates are an important form of carbon released from plants to soil. Werth and Kuzyakov observed carbon dioxide using radioactive labeling and found that 20% to 40% of the carbon fixed by plant photosynthesis is released as root exudates [35]. Organic acids are the main components of plant root exudates. Under the stress of nutrient deficiency, plant roots may be stimulated to secrete more organic acids [36]. Organic acids can be used as carbon sources by microorganisms into more complex carbon-containing compounds, and can also be adsorbed by soil and stored in soil [31]. Among the tested organic acids, only five organic acids were detected in the whole growth period of tartary buckwheat, and the changes in these five organic acids could reflect the changes in plant roots and microbial activity in the soil. The correlation between malonic acid and propionic acid was significant and both of them were detected under the same conditions, which indicated that the mechanisms of the secretion of these two organic acids by plants and microorganisms were very similar, while sterilization significantly affected the production of HF and DQ malonic acid and propionic acid. The two organic acids of DQ and HF appeared before and after sterilization at the flowering stage, respectively.

Sterilization treatment distinguished the number and type of organic acids from plants and microorganisms. For example, a decreased trend of oxalic acid was found in both of HF and DQ (DQ decreased more). It could be concluded that the two kinds of tartary buckwheat microorganisms secrete oxalic acid at the flowering stage, and the microbial secretion in DQ soil was dominant. Oxalic acid is widely distributed in plants and microorganisms and is an important metabolite in organisms. Studies had shown that oxalic acid can activate inorganic phosphorus, thereby promoting plant and microbial growth [37], and oxalic acid improved the barren tolerance of DQ. Tartaric acid has an antioxidant capacity and can effectively resist the invasion of ozone in plant seedlings [38]. Therefore, tartaric acid was detected in the soil of tartary buckwheat at the seedling stage after sterilization, while the changes of two kinds of tartary buckwheat at the flowering stage after sterilization showed that the main sources of tartaric acid of HF and DQ were plants and microorganisms, respectively. The study found that adding nitrogen to phosphorus-deficient plants reduced shoot growth and enhanced tartrate release, which may discharge excess carbon [39], while DQ tartary buckwheat showed stronger tartaric acid secretion in low-nutrient environments.

Malic acid can provide carbon sources for microbial communities, thereby stimulating the growth of microorganisms and enriching community diversity. At seedling and flowering stages, DQ and HF maintained their advantages before and after sterilization, which was consistent with other organic acids. Among all the organic acids detected, the total amount of organic acids in DQ was higher than that in HF during the growth period under untreated conditions, indicating that DQ had a stronger carbon emission capacity. After sterilization, the total amount of organic acids except oxalic acid showed that HF was greater than DQ, indicating that microorganisms were the main driving factor for carbon emission from DQ roots.

In addition, correlation analysis can reflect the correlation between the status of different organic acids and secretion. It could be found that tartaric acid had the highest correlation with other organic acids in HF and DQ, which indicated that tartaric acid played a dominant role in the secretion of tartary buckwheat roots. While malonic acid and propionic acid had the highest correlation with tartaric acid in HF, so did the tartaric acid and oxalic acid in DQ.

### 4.3. Response of Soil C-Transforming Enzymes to Sterilization

C-transforming enzyme is an important participant in the process of the soil carbon cycle. It helps organisms to absorb and utilize carbon by gradually decomposing complex carbon-containing compounds into small molecular-weight substances [40]. Increased C-transforming enzymes activity has long been considered to increase carbon emissions and soil carbon losses, but studies have also found that increased soil enzymes that decompose sugar-containing compounds reduce the use of lignin and other difficult-to-decompose carbon-containing compounds by microorganisms, thereby improving carbon stability [41]. Glucosidase Microorganisms limit the decomposition of cellulose into glucose, also known as monosaccharide enzymes [42]. In this study, sterilization increased HF but decreased the glucosidase activity of DQ during the growth period, while the content of HF was much lower than that of DQ at the flowering and maturity stages. It could be concluded that the source of glucosidase of DQ in the low-nutrient environment was mainly microorganisms, and the demand for DQ for monosaccharides was greater than that of HF.

Sucrase can hydrolyze sucrose to produce glucose and fructose, also known as disaccharidase [43]. Most sucrase is derived from plant root exudates, which can effectively reflect the respiration intensity and carbon emission efficiency of plants [44]. Sterilization increased the sucrase activity of HF and DQ at the seedling stage and flowering stage. This indicated that sterilization destroyed the ability of soil microbes to decompose carbon, and the roots could secrete more disaccharidase to meet the demand for carbon under stress without a suitable carbon source. In addition, the invertase of DQ and HF showed the opposite trend in the plant growth period under the sterilization treatment, which may indicate that DQ roots had stronger root activity in the flowering period and HF roots only showed strong growth activity in the seedling stage.

Cellulase is one of the important components of plant residues in soil carbohydrates, and also the main means of microbial decomposition and utilization of plant residues [45]. Its substrate structure is complex, and the initial hydrolysis product is cellobiose, which is eventually decomposed into glucose, so it could also name a polysaccharide enzyme [26]. Sterilization increased the cellulase activity of tartary buckwheat at the mature stage, which may be due to the strong oxidation of ozone and the decrease of plant nutrient utilization efficiency caused by sterilization, which hindered the normal growth of plants. Therefore, tartary buckwheat litter increased and released a large amount of cellulase from the residue. After sterilization, DQ cellulase increased significantly in three periods, while HF did not change the same, indicating that ozone sterilization stimulated DQ to seek more complex carbon sources to meet their own needs, which was not conducive to soil carbon storage.

DQ is more inclined to use simple carbon sources in low-nutrient environments, and microorganisms play an important role. When the participation of microorganisms was inactive, DQ will secrete more disaccharide and polysaccharide enzymes to adapt to the environment. Both organic acids and soil enzymes are both root exudates and can also interact with each other through microorganisms. Studies have shown that the release of organic acids into the soil produces a stimulating effect, which in turn promotes microbial growth and stimulates an increase in soil enzyme activity [46]. Through RDA analysis of the relationship between soil enzymes and physical and chemical factors and organic acids, it can be seen that organic acids played a positive role in β-glucosidase at the seedling stage and flowering stage, which reflected the promoting effect of organic acids on microbial growth. At the same time, cellulase activity showed a negative correlation with most organic acids, which may be because organic acids stimulated the decomposition and utilization of monosaccharide enzymes by microorganisms, thereby reducing the consumption of polysaccharides in soil, which was conducive to the stable accumulation of soil carbon.

### 4.4. Carbon Sequestration Capacity of Different Cultivars

According to the structural equation model, it can be seen that ozone sterilization indirectly affected the organic carbon sequestration of tartary buckwheat roots. Specifically, in the absence of microbial participation, the reduction of organic acid emissions led to the lack of organic carbon supplementation. At the same time, the lack of microbial decomposition and transformation of carbon-containing compounds makes tartary buckwheat turn to using more complex carbon sources. However, due to the increase in the use of sugar-containing compounds, more complex carbon is preserved, which indirectly promotes carbon accumulation. Sterilization also promoted the secretion of more refractory organic acids by plants, resulting in a decrease in pH, but promoting the stability of soil organic carbon. Therefore, after the loss of microbial participation, tartary buckwheat can still promote the accumulation of organic carbon by regulating root exudates and soil enzymes during the growth period, which also reflects the potential of tartary buckwheat in the direction of carbon sequestration. From the perspective of differences, after replacing the poor-tolerant variety of tartary buckwheat with the poor-tolerant variety, the organic acid content and soil enzyme activity increased, but the organic acid was not well preserved in the soil, which increased the loss of carbon. However, it stimulated the root to secrete more carbon-containing compounds after increasing the utilization of disaccharides and polysaccharides. At the same time, the reduction of soil water content caused by the replacement of varieties further reduced soil carbon loss. Therefore, without considering the effect of microorganisms, the carbon sequestration capacity of barren-tolerant varieties was weaker than that of barren-intolerant varieties, but barren-tolerant tartary buckwheat had greater potential in carbon sequestration capacity in practical application.

## 5. Conclusions

In the whole process of tartary buckwheat growth, Diqing plant roots have stronger carbon input capacity, and microorganisms also promote it. The microorganisms in the Heifeng tartary buckwheat roots hindered the root carbon input capacity. There were also significant differences in the combination and content of organic acids secreted by different varieties, which may be one of the reasons affecting the role of rhizosphere microorganisms. In the absence of microorganisms, the rhizosphere soil of tartary buckwheat can regulate the storage of soil organic carbon by secreting extracellular enzymes and organic acids. The structural equation model showed that to promote carbon sequestration, Heifeng tartary buckwheat needed to inhibit microbial action when planted in the barren area of Loess Plateau, while Diqing tartary buckwheat needed to use microbial-promoting agents.

## Figures and Tables

**Figure 1 ijerph-20-00959-f001:**
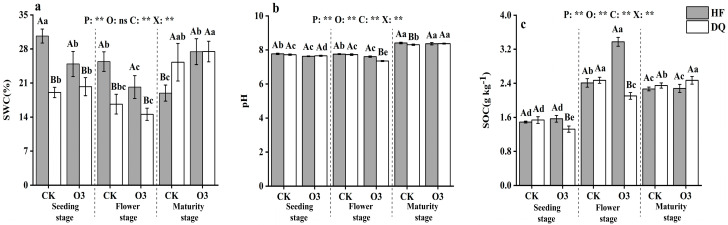
The activity of soil water content (SWC), pH, and soil organic carbon (SOC) in the rhizosphere soil of DQ and HF tartary buckwheat at seedling (**a**), flowering (**b**), and maturity stages (**c**). DQ represents low soil nutrient tolerant variety and HF represents low soil nutrient sensitive variety. Bars with different letters indicate a significant difference in the least significant difference (LSD) tests (*p* < 0.05) between different cultivars (capital letters) and treatments (small letters). P means treatment period, O means treatment effect, C means cultivar effect, and X means interactive effects of treatment, period, and cultivar. CK, Controlled experiment. O3, Ozone sterilization treatment. ** indicates a significant difference (*p* < 0.01), and ns indicates no significant difference (*p* > 0.05).

**Figure 2 ijerph-20-00959-f002:**
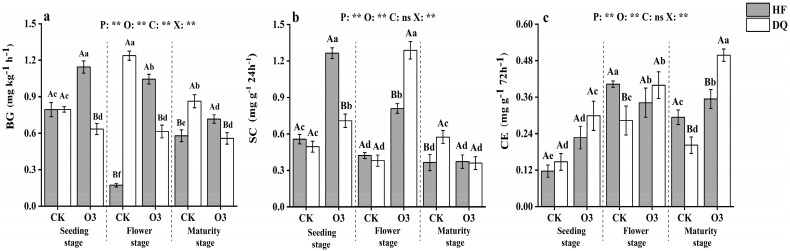
Activity of β−glucosidase, sucrase, and cellulases in rhizosphere soil of DQ and HF tartary buckwheat at seedling (**a**), flowering (**b**), and maturity stages (**c**). BG, β−glucosidase; SC, sucrase enzyme; CE, celluase enzyme. DQ represents low soil nutrient tolerant variety and HF represents low soil nutrient sensitive variety. Bars with different letters indicate a significant difference in the least significant difference (LSD) tests (*p* < 0.05) between different cultivars (capital letters) and treatments (small letters). P means samples period effects, O means germicidal treatment effects, C means variety effects, and X means the interactive effects of treatments, period, and variety. CK, Controlled experiment. O3, Ozone sterilization treatment. ** indicates a significant difference (*p* < 0.01), and ns indicates no significant difference (*p* > 0.05).

**Figure 3 ijerph-20-00959-f003:**
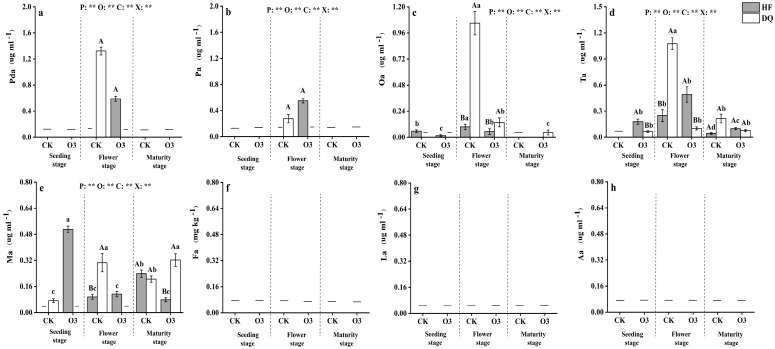
Content of organic acid in rhizosphere soil of DQ and HF tartary buckwheat at seedling, flowering, and maturity stages. (**a**) the content of malonic acid in rhizosphere soil of DQ and HF tartary buckwheat at seedling, flowering, and maturity stages. (**b**) the content of propionic acid in rhizosphere soil of DQ and HF tartary buckwheat at seedling, flowering, and maturity stages. (**c**) the content of oxalic acid in rhizosphere soil of DQ and HF tartary buckwheat at seedling, flowering, and maturity stages. (**d**) the content of tartaric acid in rhizosphere soil of DQ and HF tartary buckwheat at seedling, flowering, and maturity stages. (**e**), the content of malic acid in rhizosphere soil of DQ and HF tartary buckwheat at seedling, flowering, and maturity stages. (**f**) the content of formic acid in rhizosphere soil of DQ and HF tartary buckwheat at seedling, flowering, and maturity stages. (**g**) the content of lactic acid in rhizosphere soil of DQ and HF tartary buckwheat at seedling, flowering, and maturity stages. (**h**) the content of acetic acid in rhizosphere soil of DQ and HF tartary buckwheat at seedling, flowering, and maturity stages. DQ represents low soil nutrient tolerant variety and HF represents low soil nutrient sensitive variety. Bars with different letters indicate a significant difference in the least significant difference (LSD) tests (*p* < 0.05) between different cultivars (capital letters) and treatments (small letters). P means treatment period, O means treatment effect, C means cultivar effect, and X means interactive effects of treatment, period, and cultivar. ** indicates a significant difference (*p <* 0.01), and ns indicates no significant difference (*p >* 0.05). Pda, malonic acid; Pa, propionic acid; Oa, oxalic acid; Ta, tartaric acid; Ma, malic acid; Fa, formic acid; La, lactic acid; Aa, acetic acid.

**Figure 4 ijerph-20-00959-f004:**
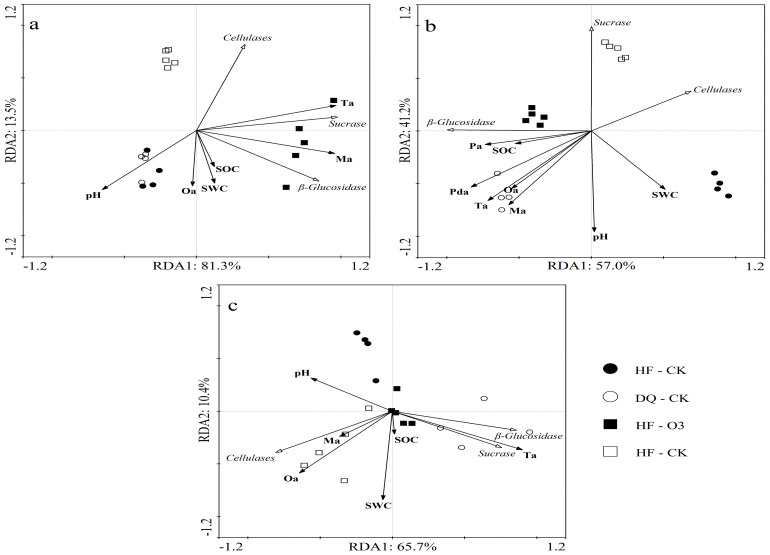
Redundancy analysis (RDA) of C-transforming enzymes, organic acid, and environmental factors at seedling (**a**), flower (**b**) and maturity(**c**) stages. The meaning of environmental factor representatives is consistent with the above figure. solid and hollow represent the treatment under HF and DQ varieties, respectively. DQ represents low soil nutrient tolerant variety and HF represents low soil nutrient sensitive variety. circle and square represent CK and O3 treatments, respectively. Pda, malonic acid; Pa, propionic acid; Oa, oxalic acid; Ta, tartaric acid; Ma, malic acid; BG, β−glucosidase; SC, sucrase enzyme; CE, celluase enzyme.

**Figure 5 ijerph-20-00959-f005:**
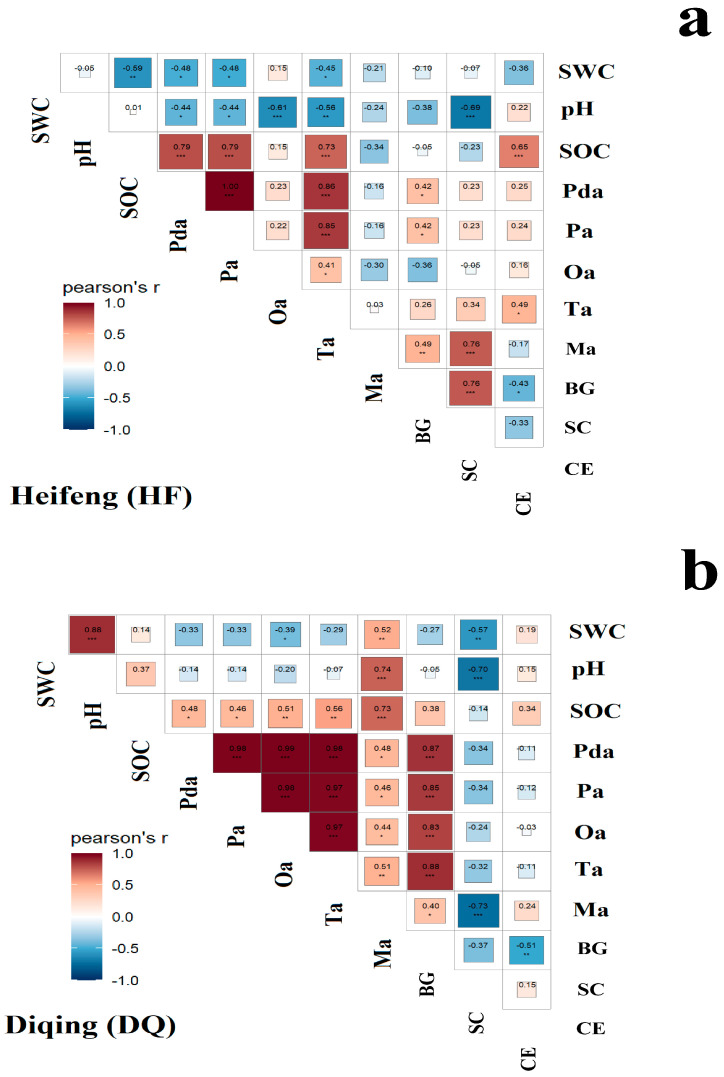
Pearson’s correlation analysis of environmental factors, organic acid, and C-transforming enzymes in HF (**a**) and DQ (**b**). DQ represents low soil nutrient tolerant variety and HF represents low soil nutrient sensitive variety. SOC, soil organic carbon; SWC, soil water content; Pda, malonic acid; Pa, propionic acid; Oa, oxalic acid; Ta, tartaric acid; Ma, malic acid; BG, β−glucosidase; SC, sucrase enzyme; CE, cellulase enzyme. ** p <* 0.05; *** p <* 0.01; **** p <* 0.001.

**Figure 6 ijerph-20-00959-f006:**
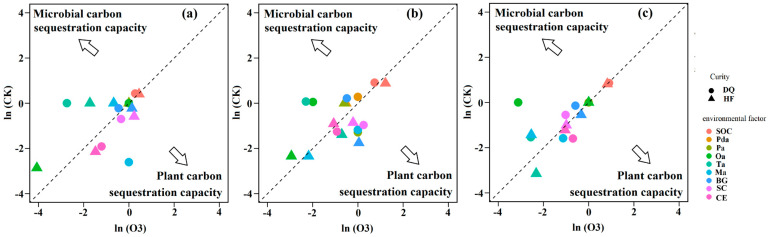
The ratio of CK treatment and O3 treatment using ln function in seedling (**a**), flower (**b**), and maturity (**c**) stages. Different colors represent different indicators, and triangles and circles represent HF (low soil nutrient-sensitive variety) and DQ (low soil nutrient tolerant variety), respectively. SOC, soil organic carbon; SWC, soil water content; Pda, malonic acid; Pa, propionic acid; Oa, oxalic acid; Ta, tartaric acid; Ma, malic acid; BG, β−glucosidase; SC, sucrase enzyme; CE, celluase enzyme.

**Figure 7 ijerph-20-00959-f007:**
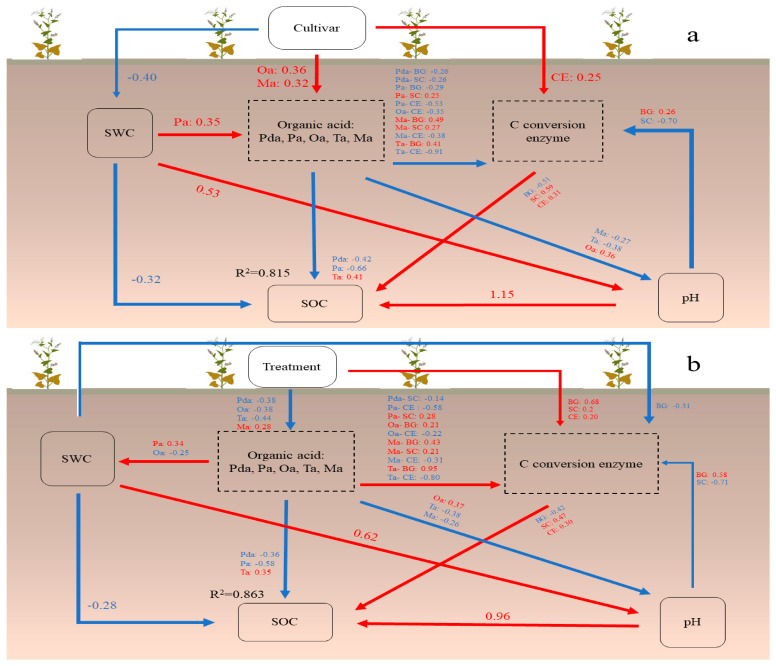
Structural equation model (SEM) of different Tartary buckwheat cultivars and treatment in (**a**) (x^2^ = 33.787, df = 29, *p* = 0.247, CFI = 0.988, GFI = 0.911, RMSEA = 0.056) and (**b**) (x^2^ = 37.805, df = 26, *p* = 0.063, CFI = 0.974, GFI = 0.907, RMSEA = 0.093). Rectangles represent observed variables, arrow thickness represents the magnitude of the path coefficient. The Red and blue lines represent positive and negative effects between the observed variables. SOC, soil organic carbon; SWC, soil water content; Pda, malonic acid; Pa, propionic acid; Oa, oxalic acid; Ta, tartaric acid; Ma, malic acid. BG, β−glucosidase; SC, sucrase enzyme; CE, celluase enzyme.

**Table 1 ijerph-20-00959-t001:** The total, direct, and indirect effects of different indicators on soil organic carbon (SOC) in the structural equation model of different cultivars.

	Cultivar	SWC	Ma	Ta	Oa	pH	Pa	Pda	CE	SC	BG
Total Effects	−0.096	−0.2	−0.386	−0.342	0.093	0.597	−0.551	−0.442	0.357	0.594	−0.512
Direct Effects	0	−0.325	0	0.415	0	1.146	−0.66	−0.419	0.357	0.594	−0.512
Indirect Effects	−0.096	0.125	−0.386	−0.757	0.093	−0.549	0.108	−0.023	0	0	0

SWC, soil water content; Pda, malonic acid; Pa, propionic acid; Oa, oxalic acid; Ta, tartaric acid; Ma, malic acid; BG, β–glucosidase; SC, sucrase enzyme; CE, cellulase enzyme.

**Table 2 ijerph-20-00959-t002:** The total, direct, and indirect effects of different indicators on soil organic carbon (SOC) in the structural equation model of different treatments.

	Treatment	SWC	Ma	Ta	Oa	pH	Pa	Pda	CE	SC	BG
Total Effects	0.163	0.093	−0.275	−0.426	−0.029	0.388	−0.594	−0.429	0.299	0.472	−0.416
Direct Effects	0	−0.278	0	0.354	0	0.962	−0.582	−0.361	0.299	0.472	−0.416
Indirect Effects	0.163	0.371	−0.275	−0.78	−0.029	−0.574	−0.012	−0.068	0	0	0

SWC, soil water content; Pda, malonic acid; Pa, propionic acid; Oa, oxalic acid; Ta, tartaric acid; Ma, malic acid; BG, β–glucosidase; SC, sucrase enzyme; CE, cellulase enzyme.

## Data Availability

Not applicable.

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
