# Peer review of "Differences in Carbon Sequestration Ability of Diverse Tartary Buckwheat Genotypes in Barren Soil Caused by Microbial Action"

_ijerph, 2023, doi:10.3390/ijerph20020959_

Round 1
Reviewer 1 Report
The manuscript entitled "Differences in Carbon Sequestration Ability of Diverse Tartary Buckwheat Genotypes in Barren Soil Caused by Microbial Action" The current study proposes to measure the carbon fixation capacity of two plant species with different capacities, in addition to relating specialized interaction with and without microorganisms ground.
The article is well written and relevant, and I consider it novel or original enough to be published in the International Journal of Environmental Research and Public Health.
The manuscript in its submitted form should be accepted with minimal corrections for publication.
Observation by section:
Introduction
Add the following data to the writing:
The area used for the production of tartary buckwheat in China.
Line 41, put point and separate.
Line 63, put bibliographical reference
Line 71, put bibliographical reference
Methodology
Line 91-92, delete and go to results
Line 102-107, Explain better the experimental design of the soil sampling (how much weight) and its transfer to the laboratory, in addition to including the coordinates and the type of crop present, as well as the days selected for the sampling.
For the greenhouse section, better explain the experimental design for taking soil samples (how much weight, at what depth was extracted per pot, weight of the fine composite sample), since it is not mentioned.
Line 120, put bibliographical reference
Results
It is recommended to put the results in the same order as described in the methodology.
It is recommended to improve the figures, since the legend is not understood, and it is confusing to interpret, improve significantly for all cases.
Line 189, put a space (P<0.05)
Line 197, put a space (P<0.05)
Because I work with Spearman's correlation and not with Pearson's, which is more accurate and is consistent with the variables presented.
Figure 4, 5 and 6 make them larger and more visible.
Discussion
Line 480, delete "In conclusion",
Conclusion
adequate
Bibliography
adequate
Author Response
Dear Editor and Reviewers:
On behalf of my co-authors, we are very grateful to you for giving us an opportunity to revise our manuscript. We appreciate you very much for your positive and constructive comments and suggestions on our manuscript entitled “Differences in Carbon Sequestration Ability of Diverse Tartary Buckwheat Genotypes in Barren Soil Caused by Microbial Action” (ID: ijerph-2107779).
Please refer to the attachment for specific information.

Reviewer 2 Report
Your manuscript entitled “Differences in Carbon Sequestration Ability of Diverse Tartary Buckwheat Genotypes in Barren Soil Caused by Microbial Action” has focused on understanding the relationship among carbon content of the soil, buckwheat genotypes and activity of microorganisms in a greenhouse experiment. Your manuscript contains more interesting information on this research topic which can help clarify the effect of the tested parameters on changes in the soil. However, there are some unclarified parameters in the manuscript.
- Certainly, microorganisms influence the development of crop plants. The microorganism ×plant interactions have also a role in changes of soil properties. What kind of microorganisms were in the collected soil samples (control)? Did you determine the genus, species of bacteria, and fungi in the soil sample? You should focus on the characterization of microorganisms.
- Two buckwheat genotypes were tested from the sowing (seedling stage) till the harvest (maturing stage) in the experiment. How did the treatments influence the growing of buckwheat genotypes (plant height, flowering time, harvesting time, the yield of individual plants, thousand kernel weight etc.)? You should detail some phenotypic data of buckwheat genotypes.
- You can demonstrate the effects of treatments (for example plant height etc.) and differences between genotypes with some photos.
- Abstract: The first sentence is unnecessary.
- The data were collected in three phenological stages (seedling, flowering and mature). However, different expressions are applied in the manuscript for the same stages (for example bleeding stage-seedling stage, flower stage-flowering stage).
- line 304-308. This part belongs to Materials and methods.
- You have to use the abbreviations consequently in the whole manuscript. It is confused in the present form (for example organic acids).
Based on the above-mentioned reasons, I can not support the publication of present form of this manuscript in IJERPH.
Author Response

(The authors gave the same response as above.)

Reviewer 3 Report
It is a very well prepared paper. A detailed methodology section, interesting results, insightful discussion, up-to-date references. My few comments do not influence the scientific value of this paper. I recommen publishing it in the IJERPH after minor revision.
General comments:
1. Figures are too small and therefore hard to read (especially descriptions).
Specific comments:
2. Lines 9-11: Please remove this sentence: „The abstract should be an objective representation of the article and it must not contain results that are not presented and substantiated in the main text and should not exaggerate the main conclusions.”
3. Lines 19: Please specify that Diqing tartary buckwheat planting has more advantages than using Heifeng tartary buckwheat.
4. Line 92: Please add information on soil texture (content of sand, silt, clay). Also, I suggest adding soil name according to Word Reference Base for Soil Classification (WRB). But it is Authors’ decision. Good that you included name according to Chinese classification. WRB nomenclature would facilitate comparison of your study with other studies in the future.
5. Lines 93-97: Please add dimensions of each pot. How many pots did you use?
6. Lines 98-107: Please use upper- and lowercases when applicable, e.g. ‘L-1’ instead of ‘L-1’.
7. Figure 1, 2, 3: Description: did you mean M means seeding stage?
8. Line 253: You should introduce here a full name of RDA, as you use the abbreviation for the first time, e.g. “The redundancy analysis (RDA)…”. Also, did you mention this analysis in the ‘Materials & methods’ section? Same with Spearman correlation analysis (line 281) and the ln function (line 304).
9. Fig. 6: Description: seedling or seeding?
10. Figure 7, Table 1 and Table 2 should be placed in the ‘Results’ section, not in the ‘Discussion’.
11. Table 1 and 2: I suggest using the same order of parameters in both tables.
Author Response

(The authors gave the same response as above.)

Round 2
Reviewer 2 Report
The manuscript can be accepted in its present form.